

# Optimising 4D Approaches to Surface Change Detection: Improving Understanding of Rockfall Magnitude-Frequency

Jack G. Williams[1], Nick J. Rosser[1], Richard J. Hardy[1], Matthew J. Brain[1], and Ashraf A. Afana[2]

[1]Department of Geography, Durham University, Lower Mountjoy, South Road, Durham, UK. DH1 3LE
[2]National Trust, Kemble Drive, Swindon, UK. SN2 2NA

*Correspondence to*: Jack G. Williams (j.g.williams@durham.ac.uk)

**Abstract.** We present a monitoring technique tailored to analysing change from near-continuously collected, high-resolution 3D data. Our aim is to fully characterise geomorphological change typified by an event magnitude frequency relationship that adheres to an inverse power law or similar. While recent advances in monitoring have enabled changes in volume across more
than seven orders of magnitude to be captured, event frequency is commonly assumed to be interchangeable with the time-averaged event numbers between successive surveys. Where events coincide, or coalesce, or where the mechanisms driving change are not spatially independent, apparent event frequency must be partially determined by survey interval.

The data reported has been obtained from a permanently installed terrestrial laser scanner, which permits an increased frequency of surveys. Surveying from a single position raises challenges, given the single viewpoint onto a complex surface
and the need for computational efficiency associated with handling a large time series of 3D data. A workflow is presented that optimises the detection of change by filtering and aligning scans to improve repeatability. An adaptation of the M3C2 algorithm is used to detect 3D change, to overcome data inconsistencies between scans. Individual rockfall geometries are then extracted and the associated volumetric errors modelled. The utility of this approach is demonstrated using a dataset of $\sim 9 \times 10^3$ surveys acquired at $\sim 1$ hour intervals over 10 months. The magnitude-frequency distribution of rockfall volumes
generated is shown to be sensitive to monitoring frequency. Using a 1 h interval between surveys, rather than 30 days, the volume contribution from small ($< 0.1$ m$^3$) rockfall increases from 67% to 98% of the total, and the number of individual rockfall observed increases by over three orders of magnitude. High frequency monitoring therefore holds considerable implications for magnitude-frequency derivatives, such as hazard return intervals and erosion rates. As such, while high frequency monitoring has potential to describe short-term controls on geomorphological change and more realistic magnitude-
frequency relationships, the assessment of longer-term erosion rates may be more suited to less frequent data collection with lower accumulative errors.

## Keywords

Magnitude-frequency, Rockfalls, 4D Monitoring, Terrestrial Laser Scanning, Point Clouds





## 1 Introduction

The erosion and transfer of mass are fundamental drivers of landscape evolution at a range of spatial and temporal scales. The processes that erode landscapes involve a broad range of event sizes, the distribution of which are commonly characterised using magnitude-frequency curves. Wolman and Miller (1960) proposed that the frequency of events that denude the Earth's
surface is log-normally distributed, and that their geomorphic effectiveness (the product of magnitude and frequency) is greatest for the frequent, moderately sized events. This concept has been widely applied both to study the geomorphic efficacy of rivers (Wolman and Gerson, 1978; Hooke, 1980; Nash, 1994; Gintz et al., 1996) and the characteristics of landslides (Hovius et al., 1997; 2000; Dussauge-Peisser et al., 2002; Turcotte et al., 2002; Dussauge et al., 2003; Malamud et al., 2004; Guthrie and Evans, 2007; Li et al., 2016) using inverse power law distributions or similar.

The exponent of the inverse power law describes the proportional contribution of increasingly small events. However, many landslide volume distributions have been characterised by a decrease in the frequency density of the smallest events in log magnitude-log frequency space, known as a 'rollover', (Malamud et al., 2004). At this point, the inverse power law breaks down, and so alternative distributions such as the double Pareto (Stark and Hovius, 2001; Guzzetti et al., 2002) or inverse Gamma (Malamud et al., 2004; Guzzetti et al., 2005) have been drawn upon to model observations. Explanations for this
rollover have been widely considered, and include mechanical differences and physically based minimum possible event sizes (Pelletier et al., 1997; Guzzetti et al., 2002; Guthrie and Evans, 2004), or censoring of the smallest events by the resolution or frequency of monitoring (Lim et al., 2010). For rockfall, Malamud et al. (2004) hypothesised that a rollover may not occur due to rock mass fragmentation.

The duration of monitoring relative to the return period of all possible event sizes determines the likelihood of detecting
changes that are representative of how a landform evolves over longer timescales. The completeness of an event inventory is also a function of the smallest event size that can be detected, and the temporal frequency of monitoring compared to the rate at which such small events occur. Abellán et al. (2014) suggest that the spatial resolution of rockfall monitoring should be sufficient to discretise the smallest events in a magnitude-frequency distribution, and that the recording frequency should fall below the timescale in which superimposition and coalescence may occur. In practice, defining this timescale *a priori* is
challenging and requires the ability to monitor the rock face over a sustained period in (near) real-time. For rockfall, high-resolution monitoring also shows evolution of failures through time, with event sequences and patterns related to the incremental growth of scars (Rosser et al., 2007; 2013; Stock et al., 2011; Kromer et al., 2015; Rohmer and Dewez, 2015; Royán et al., 2015). Barlow et al. (2012) showed that a monitoring interval of 19 months underestimated the frequency distribution of small rockfall events, which coalesced into or were superimposed by larger rockfall. Treating rockfall as
spatially and temporally independent is therefore problematic, as is experienced in other types of landform change. For example, Milan et al. (2007) found an increase in erosion and deposition volumes within a proglacial river channel when monitored using daily terrestrial laser scan (TLS) surveys as opposed to surveys separated by eight days. This was attributed



to the temporal length scales of discharge and sediment supply, with return periods of less than eight days. The influence of monitoring or sampling interval on measured process rates are more widely considered in the Sadler effect, in which sediment accumulation rates observed in stratigraphic sections exhibit a negative power law dependence on measurement interval (Sadler, 1981; Wilkinson, 2015). Importantly, without higher frequency monitoring, in settings that change little but often, the

ability to capture true magnitude and frequency is therefore subject to an unknown degree of superimposition and coalescence, and temporal coincidence.

The improvements in temporal resolution from (semi-)permanent monitoring installations (e.g. Kromer et al., 2017) are weighed against a series of compromises in the quality of data generated. This includes artefacts that arise from scanning from a single position onto a complex surface, which results in occlusion, leaving 'holes' in the point cloud in areas invisible to the

scanner. Further, as laser scanners never measure exactly the same point twice (Hodge et al., 2009), the perimeter of these holes will move between successive point clouds, despite no movement of the instrument. This effect is exacerbated on surfaces with high relief due to the averaging of multiple range measurements within a single laser footprint (Lichti et al., 2005; Hodge et al., 2009). Scan lines in most laser scanners result in non-uniformly distributed data, with heterogeneity often at a scale and orientation comparable to surface structure, leading to aliasing that is also inconsistent (Lichti and Jamtsho, 2006). The

influence of these combined effects is exaggerated if the scanner view is oblique to the surface, which may not be uncommon when siting a semi-permanent instrument.

While studies that draw upon more frequent 3D monitoring have highlighted the benefits of capturing the frequency of small events, an inevitable consequence is the error in change detection between each sequential dataset that accumulates in proportion to the total number of scans (Brasington et al., 2000). The result is greater levels of uncertainty in the total volume

of change estimates at higher survey frequencies, which increases with the duration length of the monitoring campaign. To handle such errors, Wheaton et al. (2010) proposed a fuzzy inference method to model spatially variable digital elevation model (DEM) error prior to planar surface change detection. This approach draws upon a three-input rule system relating to the point cloud slope, point density and 3D point quality from GPS. This is applied cell-by-cell to individual component DEMs. A second approach modifies 'DEMs of Difference' (DoDs) based on the coherence of erosion and deposition surrounding

each cell, which is contingent upon unidirectional change at scales greater than the cell size. For rockfall monitoring, however, neighbouring cells that exhibit movement in opposite directions may represent a number of important failure mechanisms at scales comparable to the cell size, such as the loss of material (erosion) from the surface of a forward-creeping wedge failure (deposition). The method described in this study derives change in 3D using $\sim 10^3$ individual 3D scans, with each comprising $> 10^6$ points. This is used to describe a full rockfall magnitude-frequency distribution, and so the approach is optimised for

handling large $(10^3 – 10^4)$ numbers of high resolution 3D scans, critically without user intervention.





## 2 System description

Date is presented from a monitored coastal cliff in North Yorkshire, UK. The rock cliff, located at East Cliff, Whitby, is near-vertical reaching ~ 60 m in height and is actively eroding. The erosion of this coast has previously been monitored and averages ~ 0.1 m a$^{-1}$ (Rosser et al., 2005; 2007; Miller, 2007). Typical rockfall include small-scale joint defined wedges, and larger-

scale failures released via rock bridge breakage, which can be inferred from the exposed fresh fracture surfaces visible after failure. Rockfall have been measured up to $2.5 \times 10^3$ m$^3$, but the volume loss is dominated by smaller scale rockfall with median volumes approaching $1.0 \times 10^{-3}$ m$^3$ (Rosser et al., 2013). The slope monitoring system presented surveys the cliff using a remotely controlled Riegl VZ-1000 laser scanner, housed inside the former lantern room at the top of East Pier lighthouse (Fig. 1), ~ 350 m seaward of the cliff face.

The viewpoint of the scanner results in some loss of spatial continuity in surface measurement due to occlusion, as a result of surface relief and the high incidence angle of parts of the rock face relative to the scanner (Fig. 2b). The closest point on the cliff is 342 m from the scanner with an incidence angle onto the strike of the cliff of ~ 25°. The furthest monitored point is 533 m from the scanner with an incidence angle of ~ 42°. Range correction for atmospheric effects and precise point cloud alignment is automatically conducted every 3 h using very high resolution scanning ($5.0 \times 10^{-4}$ m point spacing) of six fixed

square 0.25 m$^2$ control targets, the precise relative positions (± 0.005 m) of which are known from a total station survey. Atmospheric range correction, derived from comparing scanned to surveyed target ranges typically scales range measurements by ~ $1.0 \times 10^{-5}$ with minimal diurnal fluctuation, and is therefore largely inconsequential at this site, but would be significant for locations with greater ranges or areas subject to more extreme atmospheric conditions. The TLS survey was managed using SiteMonitor4D (3D Laser Mapping Ltd.), which schedules scans, manages the atmospheric correction, and applies an affine

rigid-body rotation matrix to compensate for tilt and yaw in the scanner position based upon the scanned control target positions in real-time. The reported dataset has been collected using this setup between March 2015 and December 2015, totalling ~ 9 000 scans with ~ 1 h intervals between surveys. Gaps in the dataset arise from system outages, and are excluded from the analysis.

## 3 Optimising point clouds for change detection

While the specification of successive scans is identical, and given that neither the instrument position nor the surface have changed, individual point clouds differ due to the inherent uncertainties in point clouds described above. The data processing method described involves point cloud filtering, alignment, 3D change detection, interpolation of the point clouds of change, and classification of this change into 2.5D datasets from which 3D change geometry can be analysed. While this paper does not seek to create a 'real-time' system, the approach described is optimised for computational efficiency to allow data to be

processed at a rate that is at least as quick as collection.





## 3.1 Point Cloud Filtering

Scan data collected outside of the area of interest (AOI) can provide useful information for scan-to-scan registration, particularly if it covers a wider geographical area, has a distribution that is less planar than the AOI itself, or where a large portion of the AOI is undergoing deformation. Here, given that the control target network provides positional control, the

spatial extent of the AOI is clipped (Kemeny and Turner, 2008). For a small number of scans (< 20), this can be feasible to undertake manually. A cuboidal bounding box is applied automatically here, which typically reduces the raw dataset from $\sim 1.9 \times 10^{6}$ to $1.1 \times 10^{6}$ points.

Repeatability in change detection is dependent on point clouds that consistently describe the monitored surface. Optimising a point cloud for change detection involves removing points on features that cannot be consistently measured, such as edges and

vegetation. A morphological and radiometric filter is applied to remove points with high positional uncertainties in a consistent manner. To detect the presence of edges in the point cloud, neighbouring points within a fixed radius of each (query) point, $q$, are identified and the central position of the neighbourhood points, $CoG$, calculated. The 3D Euclidean distance, $ED$, between $q$ and the $CoG$ is then calculated. For a point at an edge, $CoG$ tends away from the query point. The distance $ED$ is therefore larger for query points that lie closer to an edge. Applying a threshold to $ED$ needs to account for the varying point density

across the cloud as in regions of low point density $ED$ will always be larger. The value $EH$ assigned to each point is therefore reported as a ratio of the distance $ED$ to the number of points in a spherical domain centred on each point:

$$EH = \frac{ED}{k} \tag{1}$$

where $k$ is the number of neighbouring points. Figure 3 demonstrates the role played by points with high $EH$ values in

increasing the uncertainty in change detection between sequential scans of an identical surface where the 3D distance estimation between clouds is used as a proxy for the level of measurement uncertainty. Points that exhibit higher EH values are identified by a higher 3D distance between clouds. The threshold applied here, $5 \times 10^{-4}$, removed 5% of points that account for uncertainties > 0.5 m (Fig. 3a). This also helps to delineate areas of occlusion. The point density, $k$, is used to filter spurious 'floating points' in the dataset (for example birds or dust). $k$ values < 4, the minimum number to accurately define a centroid

with an associated error, were removed.

A limitation of many laser scanners is the inability to quantify the accuracy of each range measurement, and with the common absence of better data, assessing reliability is challenging. In most TLS systems used for rock slope monitoring, range is estimated using the time of flight of a laser, where time is stamped based upon some characteristic of the measured reflection (intensity gate, maximum intensity amplitude), which varies between scanners. Some systems have the ability to capture the



full energy-time distribution of the reflection. This provides a means of estimating the relative quality of recorded measurements as a function of either the number of separate reflections from a single pulse, the incidence angle of the laser beam with respect to the surface (the elongation through time of the reflected pulse relative to the emitted pulse), and the reflectance intensity (the integral of the reflection energy-time distribution). Within this approach, the characteristics of the

returned signal to remove vegetation (multiple returns per pulse) and edges (elongated reflections) are used to increase the consistency between successive point clouds.

The Riegl VZ-1000 TLS, with 'waveform' capture, records the intensity of each returned signal at $2.01 \times 10^{-9}$ s intervals, providing 15-70 amplitude measurements per pulse. The energy of the received laser pulse structure depends on the spatial and temporal energy distribution of the emitted pulse, which are modified by the geometric and reflectance properties of the

target surface (Stilla and Jutzi, 2008). The 'deviation', $\delta$, of the waveform describes the change in shape of the received waveform relative to a modelled (emitted) Gaussian energy-time distribution according to:

$$\delta = \sum_{i=1}^{N} |s_i - p_i| \qquad (2)$$

where $N$ is the observations in pulse $s_i$, compared to the reference values, $p_i$. $\delta = 0$ represents identical emitted and received

waveforms, as would be expected from a nadir oriented planar specular surface. $\delta$ is less sensitive to target range than incidence angle. The former is responsible for increasing positional uncertainty by spreading the footprint over a larger area as it intersects an off-nadir surface, reducing the energy reflected, and increasing the period of time over which the backscattered pulse is returned to the sensor (Soudarissanane et al., 2011; Hartzell et al., 2015; Telling et al., 2017).

In Figure 4a, the point-to-point differences between non-filtered data with no physical change show point position uncertainty.

Mean absolute change for points classified by $\delta$ shows that apparent change is only $\sim 0.02 – 0.03$ m for points where $\delta \leq 25$. Conversely, for points where $\delta > \sim 25$ exhibit more significant scatter, often approaching two to three times the level of uncertainty in the whole cloud. Removing points with $\delta > 25$ (Fig. 4b) retains 98% of points, which accounted for a standard deviation of error between point clouds of 0.18 m prior to removal. Similar to the edge filter, removing only those points associated with high levels of uncertainty removes artefacts that are often on the periphery of the point cloud but, if not

removed, hold a significant influence on the overall repeatability of change detection. The sequence with which these filters are applied has little bearing on the outcome and the subset of points removed by each has common members. When combined and applied to the dataset in this study, the filters described above reduced the standard deviation of change measurements between two stable point clouds from 0.078 m to 0.055 m, thereby lowering the LoD that could be applied during rockfall or deformation identification by $\sim 30\%$.



In addition to the filtering of individual points considered as unreliable, entire surveys required removal from the overall inventory due to inclement weather conditions. In conventional monitoring with no scan schedule automation, scans that are partially or fully obscured due to inclement weather conditions, such as rain or fog, are manually removed and/or repeated. Due to the frequency and duration of near constant scanning, many scans will either be partially or fully obscured. Scans that

are entirely obscured can be removed automatically with relative ease. However, unobscured areas in partial scans still allow some accurate change detection and rockfall identification, and so may be valuable to retain. Given that change is detected between scan pairs, however, it is critical that these partial scans are removed prior to change detection, and remain unused. Figure 5 describes a scenario in which a rockfall occurs between 12.00 and 12.30 during adverse weather, which partially obscures the impending scan at 12.30. While some areas of this scan allow accurate change detection of the surface, if the

rockfall occurs in an obscured area, it can be omitted from the inventory entirely. However, if surfaces are compared between 12.00 and the following scan at 13.00, with both captured during fair conditions, the rockfall will be observed and included in the inventory.

Given the variability in the persistence of inclement weather during a scan, a threshold based on the number of points is unsuited to identifying partial scans. At present, no automated method for detecting partial scans has been developed, though

the removal of any scan that coincides with measured rainfall may represent a first step. Here, the point distribution was manually examined by creating a video of every point cloud prior to reanalysis of the dataset. While the maximum possible number of change detections was 8 986, these were reduced to 8 596 as a result of poor weather conditions and finally to 8 270 as a result of partial scan removal. The reduction in the number of scans, however, has a direct impact on the time interval between scans and hence deformation analysis prior to failures that occur during bad weather, which may result in some of the

most active periods of rockfall.

### 3.2 Precise Alignment

It is assumed here that successive point clouds are approximately but not perfectly aligned, given the fixed scanner position. Once filtered, points clouds are automatically registered to a reference point cloud to improve alignment. As discussed by Abellán et al. (2014), aligning point clouds can be undertaken using common surveyed and modelled targets combined with

measured global coordinates (Teza et al., 2007; Olsen et al., 2009), feature-based registration based on the planarity and curvature of surfaces (e.g. Besl and Jain, 1988; Belton and Lichti, 2006; Rabbani et al., 2006), and point-to-point and point-to-surface methods, which use iterative closest point (ICP) alignment to progressively reduce the distance between two clouds (Besl and McKay, 1992; Chen and Medioni, 1992; Zhang, 1994). The accuracy of alignment is one of the key sources of error when detecting change between two point clouds (Teza et al., 2007). Registration of datasets into a global system has a

significant impact on data file sizes due to multi-digit coordinates, which becomes problematic with large numbers of large point clouds. In this approach, a local coordinate system is retained and an ICP registration applied using MATLAB®'s pcregrigid function (Besl and McKay, 1992; Chen and Medioni, 1992). This method searches for the closest point in the

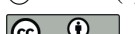



reference scan for each point in the moving scan and estimates the combination of rigid rotation and translation that best aligns them (Mitra et al., 2004). Hofer (2003) showed that for point clouds that are approximately aligned, as here, minimising the point-to-plane distance provided the best estimate of convergence. Given that the raw point cloud data commonly has systematic structure (vertical or horizontal scan lines), and where the AOI is approximately planar (a ≈ 2D rock face rather

than a fully ≈ 3D scene), the success of the ICP can be dominated by aligning structure in the data, rather than the macro-scale cliff geometry. Point-to-point minimisation in ICP was therefore found to be less effective than point-to-plane alignment in this instance, using point clouds down-sampled to a fixed 0.25 m point spacing.

There is no established protocol for choosing the ideal reference scan for alignment. This reference scan may be the first available scan, a later single scan, or an average of a subset of previous scans. Schürch et al. (2011) aligned a series of three

scans to the previous scan in a sequence, rather than to the first of the monitoring campaign, in order to gain more precise change estimates between successive surveys, at the cost of the overall positional accuracy. This procedure is advantageous as it ensures that the shape of a rapidly deforming or changing surface can be matched to the previous survey, rather than one captured considerably earlier. Here, the series of scans is aligned to the first survey. This is undertaken because ICP alignment minimises the point-to-plane distance of down sampled point clouds, such that individual (small relative to the AOI) rockfall

events do not impact upon the overall success of the alignment. Second, even with low alignment errors between scan pairs, the potential for the point clouds to drift over time increases with the number of scans sequentially aligned. While Schürch et al. (2011) assessed pairwise change between scans, high frequency data allows for change detection to be conducted over multiple intervals that the time-series enables, so aligning all scans with respect to each other is important. Third, interpolation of the point clouds of change into a 2 or 2.5D raster is simplified when points consistently occupy similar positions, which

enables the location of change to be in single locations to be analysed through time. Finally, when processing a time-series of scans, gains in efficiency are possible by creating a single octree structure using the reference scan, to which subsequent clouds can be assigned. The segmentation of each point cloud into an octree structure is therefore required only once using this approach. Here, the point clouds are segmented to perform change detection, with each point assigned a $3 \times n$ bit code, where $n$ is the maximum octree level (Frisken and Perry, 2002; Girardeau-Montaut et al., 2005; Jaboyedoff et al., 2007; 2009; Elsberg

et al. 2011; 2013; Hornung et al., 2013). For subsequent operations, such as normal vector estimation, points from both the individual octree cube and the surrounding 26 cubes are used. The subdivision level at which normal estimation and change detection is performed therefore influences only the computation time, and not the result (Girardeau-Montaut et al., 2005).

Although the mean offset between unregistered point clouds at these sites was 0.51 m, the alignment was improved using ICP by the approach described (0.0053 m). Over all time-scales in the dataset, an average registration error of 0.005 m between

any pair of point clouds was obtained ($n$ to $n+x$, where $1 > x > 8\,987$). Considerable alteration to the surface topography, via the occurrence of a single large event or the continued spalling of material over time, would require new reference scans to be assigned over shorter timescales.





### 3.3 Normal estimation

The distance between successive clouds is measured along the normal vector of each point in the cloud. Accurate estimation of each normal vector is critical in determining the magnitude and direction of change and should be derived from an

appropriately sized neighbourhood of points that adds topological context (Riquelme et al., 2014). In order to calculate the normal direction of each neighbourhood, a tangent plane must be fitted to every point and its neighbours, with each being considered as a potential plane subset. Using eigenvectors calculated from principal component analysis, the eigenvector $v_3$ with the smallest associated eigenvalue is orthogonal to the plane, and therefore defines the normal (Hoppe et al., 1992). It so follows that the plane minimises the sum of squared distances to the neighbours of query point, $p$:

$$(p_i - \bar{p}) \cdot v_3 = 0 \tag{3}$$

and passes through the centroid, $\bar{p}$:

$$\bar{p} = \frac{\sum_{i=1}^{r} p_i}{r} \tag{4}$$

where $r$ is the number of neighbours in the neighbourhood, and $p_i$ represents the Cartesian coordinates of each point within the neighbourhood (Pauly et al., 2002). The identification of a local surface normal using the third eigenvector is the equivalent to forming a total least squares fitting plane. However, in a total least squares fitting the entities in the covariance matrix are not divided by $k$, and the smallest eigenvalue is equal to the sum of the residuals squared (Pauly et al., 2002; Belton and Lichti, 2006).

The neighbourhood size strongly determines the direction of surface normals (Mitra and Nguyen, 2003; Lalonde et al., 2005; Bae et al., 2009; Lague et al., 2013; Riquelme et al., 2014). If the size of the neighbourhood is below the scale of surface roughness, the resulting normals will fluctuate in direction and are less likely to be consistent between successive point clouds. Lague et al. (2013) selected the scale at which the neighbourhood of points could best be approximated by a plane. In particular, the neighbourhood should be allowed to vary in size to accommodate non-uniform point distributions and variations in point

density, producing a planar surface from which to estimate the normal. Riquelme et al. (2014) showed that variability in the strike and dip of a rock face discontinuity occurred when the number of neighbourhood points, $k$, fell below 15, whereas values of $k > 30$ over-smoothed adjacent surfaces; $15 < k < 30$ was therefore recommended. Here, by varying the size of the





neighbourhood for each point between $0.1 - 2.5$ m, the radius that produced the most planar surface is identified. An example of this is shown in Fig. 6a, with Fig. 6b illustrating surface planarity across East Cliff. This shows a clear similarity to the distribution of point density, such that the search radius is increased in regions of low point density. Importantly, identifying the optimum neighbourhood radius for $10^3 - 10^4$ point clouds adds considerable computational cost in processing. As a

5    compromise, the neighbourhood radius of each point is made equal to the distance to the closest point in the reference cloud in Fig. 6a. Notably, the normal for each point estimated uses the second cloud, such that change is accurately measured along the normal of a planar, post-failure surface, rather than the yet-to-fail surface (Fig. 7).

The sign ambiguity of each vector is also corrected (Mitra and Nguyen, 2003; Ioannou et al., 2012). This can typically be resolved using the position of the query point, $q$, relative to the sensor position, $s$, by:

$$\hat{s} = [X_s, Y_s, Z_s] - [X_q, Y_q, Z_q] \tag{5}$$

$$\text{In } \mathbb{R}^3: \alpha = arctan(\|\hat{s} \times \hat{n}\|_2) \tag{6}$$

where $\times$ denotes the vector cross product and $\|$ denotes the Euclidean norm of the cross product. $\alpha$ denotes the angle between the unit normal vector $\hat{n}$ at q and the vector between $q$ and $s$, $\hat{s}$. If $\alpha > \frac{\pi}{2}$ or $\alpha < -\frac{\pi}{2}$, i.e. if the angle between the direction of the normal vector and the vector between the surface and the sensor is not within $\pm\ 90°$, the normal direction $\hat{n}$ is reversed:

$$\hat{n}_{rev}\langle u, v, w \rangle = \hat{n}\langle -u, -v, -w \rangle \tag{7}$$

In order to minimise the computation time required to apply Eq. 5 and Eq. 6, the axis orthogonal to the surface is introduced as a string of either 'X' or 'Y'. With this information, the relevant component of the unit vector is used to determine whether the vector should be reversed or not. For example, if the approximate range in a near-nadir point cloud is measured along the

20    y-axis and the vector $\hat{n} = \langle -u, +v, -w \rangle$, then $\hat{n}$ is directed into the surface and should be reversed using Eq. 7. For each normal, this can provide a ~ 50% reduction in the time taken for sign correction.

### 3.4 Change Detection

The distance calculation used is based upon the structure of the M3C2 algorithm, developed by Lague et al. (2013). We describe below a modification incorporated to improve the overall accuracy of change detection and to streamline the workflow when

25    applied to large time series scan datasets. Once the normal vector is estimated, a bounding cylinder with a user-defined radius



is created along the normal running through the query point. In order to enforce the boundaries of this cylinder, the orthogonal

distance between every point within the current and neighbouring 26 octree cubes and the normal vector was estimated:

$$\hat{d} = [X_n, Y_n, Z_n] - [X_p, Y_p, Z_p] \tag{8}$$

5   where $\hat{d}$ is a vector that connects each neighbour point $p$ to a point on the normal vector $\hat{n}$, such as the query point, $q$. The

projection of each point onto the normal $P$ is therefore:

$$P = q \times \hat{d}, \text{ or} \tag{9a}$$

$$P = q + \left(\frac{d \cdot \hat{n}}{\hat{n} \cdot \hat{n}}\right) \times \hat{n} \tag{9b}$$

and the orthogonal distance is:

$$d_{orth} = \sqrt{(X_n - X_P)^2 + (Y_n - Y_P)^2 + (Z_n - Z_n)^2} \tag{10}$$

Given that the position of each neighbouring point and its orthogonal distance to the normal vector are known, the cylinder

boundaries can be enforced using the user-defined cylinder radius, $r$, retaining only points where $d_{orth} \leq r$. Once the points $c$

in the cylinder are isolated for both point clouds, the mean point $CP$ is estimated by:

$$CP = \left(\frac{\sum_{i=1}^{n} x}{c}, \frac{\sum_{i=1}^{n} y}{c}, \frac{\sum_{i=1}^{n} z}{c}, \right) \tag{11}$$

and projected onto the normal vector using Eq. 9a and Eq. 9b. The mean points of each sub-cloud are subtracted to give a

distance vector, $\hat{v}$:

$$\hat{v} = CP_2 - CP_1 \tag{12}$$



If the vector of change is along the direction of the normal vector (forward movement), the dot product of both vectors is > 0. If the vector of change is counter to the normal direction (backward movement), the dot product is < 0 and the vector is inverted.

M3C2 imposes a user-defined maximum cylinder length to decrease processing times. Cylinder length is critically important for determining the accuracy of change estimation, particularly at topographic edges within the point cloud. As described, edges are likely to be more prevalent in point clouds collected from single or off-nadir viewpoints. A method to reduce the effect of edge change uncertainty in change detection is therefore required. In Fig. 8, the influence of the choice of cylinder length is illustrated with respect to a jointed rock mass surface. The plots illustrate variation in measured change for a single point. When the cylinder extends 0.25 m in both directions, only points from this surface are included in the cylinder; as such, the centroid positions of each point cloud are both fitted onto that surface. The measured distance for this point, the distance between the two centroids is +0.0011 m. With a cylinder extending ± 0.50 m, points that lie between surfaces are included in the change detection. Given that the distribution of points is rarely consistent between point clouds, the position of the centroid of each neighbourhood differs considerably from the centroids estimated using a shorter cylinder and the resulting change estimate is -0.1460 m. At a length of ±10 m, the cylinder intersects multiple surfaces and the centroid positions are averaged between these surfaces. The inclusion of a greater number of points over a wider area increases the similarity of the mean position in both point clouds, but the resulting vector of change is +0.0938 m; a difference of 0.24 m from the 0.50 m cylinder length and significantly higher than the true change estimate. To address this, a distance along the normal with variable cylinder length for each point is used, referred to as DAN VCL. The approach begins with a cylinder that extends ± 0.10 m. If fewer than four points are found, the minimum number to estimate a centroid, the cylinder extends. This process is recursive and accepts a user-defined range of cylinder lengths. The method significantly lowered the LoD (0.03 m) compared to that achieved using DoDs (1.04 m), created using the same pairs of point clouds rasterised at 0.25 m with each pixel containing > 1 point. In this specific instance, the LoD shows a five-fold improvement relative to the M3C2 algorithm applied using the same normal and fixed cylinder radius (LoD = 0.165 m).

The cylinder radius determines the degree of spatial averaging over change measurements and, as such, should be informed by the type and scale of movements under investigation. In theory, the smaller the radius, the finer the spatial detail that can be established. However, this comes with a compromise in that the increase in accuracy by accounting for neighbouring points is reduced, the likelihood of intersecting points in the second point cloud is reduced, and the statistical significance of calculations is reduced by only drawing on a small number of points. Lague et al. (2013) suggest a minimum of 20 points should be included within the cylinder for each point cloud. Here, cylinder radii between 0.15 m and 2.00 m were applied to scan pairs in which rockfall had and had not occurred. For scans in which no change occurred, the standard deviation of differences across the point cloud was greatest for the 0.15 m radius (0.018 m), which approaches the point spacing, but decreased and stabilised at a radius of 0.25 m (0.013 m). For scan pairs in which rockfall had occurred, the size and shape of rockfall was contrasted with the results of varying cylinder radii and a Hausdorff distance estimation. The Hausdorff distance





measure itself is influenced considerably by the scan line spacing and the local point density but, for this purpose, it provides

an indication of rockfall geometry with the smallest degree of spatial averaging. As the cylinder radius increases, the difference

in shape relative to the Hausdorff approximation also increases. While a radius of 0.15 m best approximates the size and shape

of the rockfall, this value is too close to the scan line spacing at all but the most proximal regions of the cliff face. A search

radius of 0.25 m was therefore selected, providing similar rockfall geometries to the 0.15 m. This emphasises the potential to

apply a variable cylinder radius across the point cloud based upon local point density, which could be helpful in future research.

## 3.5 Extracting discrete changes, and quantifying volumetric error

The delineation of areas of geomorphic change, here rockfall, involves masking regions of change that exceed a hard threshold

at the level of detection (LoD), that is either estimated locally (e.g. Wheaton et al., 2010; Lague et al., 2013) or across the

entire point cloud (e.g. Abellán et al., 2009). A single LoD was identified between scan pairs in which no rockfall occurred as

two standard deviations of the 3D change, after Abellán et al. (2009). This was comparable to the LoD recorded for every scan

pair in the dataset; hence, the maximum-recorded LoD was applied to all scan pairs in the dataset. Similar to Kromer et al.

(2017), these change estimates are assumed to include the registration error, which is reduced through range correction using

fine-scanned targets and through ICP. The resulting LoD was used to threshold 2.5D rasters of the 3D change data, created by

linear interpolation of change values across the $x$-$z$ plane. The images produced included consistently located holes (no data)

due to occlusion, which were identified and masked. Pixels that consistently exceeded the LoD within the first 100 point

clouds, including in a non-systematic manner (e.g. both forward and backward movement), were also masked. This prevented

several (predominantly single) pixels of noise from being identified as detachments.

Once a change image is thresholded according to the LoD, the volume of each erosion event, $V_E$, can be calculated as:

$$V_E = \sum_{i=1}^{N} d_i \times A_c \qquad (13)$$

where $N$ is the number of cells that delimit the event, $d_i$ is the depth of change in cell $i$ and $A_C$ is the cell area. Previous

approaches have ignored cells with a depth change below the instrument precision and assumed that erosion events with an

aerial extent $< A_C$ cannot be detected but often fail to quantify uncertainty in volume estimates derived using Eq. 13 for rockfall

with areas greater than $A_C$ (e.g. Dussauge et al., 2003; Rosser et al., 2005; Abellán et al., 2006). Basic assumptions about how

uncertainty in aerial extent propagates into volumetric uncertainty are needed, in particular for failures of varying geometry.

This is of critical importance considering the relatively low spatial resolution of raster cells (here 0.15 m) relative to the





accuracy of the change in depth within pixels recorded by TLS (here 1 in 10 000 to 1 in 100 000). Assuming any cell that lies on the boundary of an area of change can contain any fraction ($> 0$ and $< 1$) of true change, the maximum area of change $A_{E\_max}$ is:

$$A_{E\_max} = A_C \times N \tag{14}$$

In reality, Eq. 14 represents the largest possible area because the likelihood that border cells are entirely covered by the true change is small. Conversely, the theoretical minimum area $A_{E\_min}$ approaches:

$$A_{E\_min} = A_C(N - N_b) \tag{15}$$

10 where $N_b$ is the number of boundary cells. The maximum range in uncertainty associated with the area estimate of the area of change is then:

$$A_{maxerror} = A_{E\_max} - A_{E\_min} \tag{16}$$

This value can be applied as a threshold to the rockfall inventory, such that failure areas above $A_{maxerror}$ are removed. This

15 threshold, however, represents the maximum possible error associated with the rockfall area. Jahne (2000) defined the variance $\sigma_x{}^2$ of the position of a single point in an image (cell), introduced by the cell size $d_x$, as:

$$\sigma_x{}^2 = \frac{1}{\Delta x} \int_{x_n - \Delta x/2}^{x_n + \Delta x/2} (x - x_n)^2 dx = \frac{(\Delta x)^2}{12} \tag{17}$$

assuming a constant probability density function within the cell area, i.e. all positions are equally probable. The standard

20 deviation $\sigma_x$ is approximately $\frac{1}{\sqrt{12}} \approx 0.3$ times the cell size. Therefore, to accommodate for uncertainty in the position of the area of change within each boundary cell as a function of cell size, $2\sigma$ can be used as a threshold as follows:



$$A_{E\_max} = A_C \left( N + \frac{1}{\sqrt{12}} N_b \right) \tag{18a}$$

$$A_{E\_min} = A_C \left( N - \frac{1}{\sqrt{12}} N_b \right) \tag{18b}$$

$$A_{error} = A_{E\_max} - A_{E\_min} \tag{18c}$$

The volumetric error is hence:

$$V_{error} = \sum_{i=1}^{N_b} d_i \times \frac{2}{\sqrt{12}} A_c \tag{19}$$

5  Eq. 19 shows that the number of border cells relative to the total number of cells within the area of change is critical in determining the net volumetric error. A higher ratio of border cells to the total number of cells results in a greater proportional area (and hence volume) error. While this volumetric error assessment is applied to rasters of 3D-derived change, its use also extends to extraction of discrete events from DoDs.

## 4 Application to rockfall from an actively failing rock slope

The pairwise change detection method described above was applied to a near-continuous monitoring dataset collected at East Cliff, Whitby. In total, 8 987 point clouds were collected and processed to generate an inventory of 3D rockfall geometries. The LoD was derived for every sequential scan to ensure that no increase in registration or epistemic errors developed through the monitoring period. This value lay consistently between $0.01 - 0.03$ m. The maximum LoD, 0.03 m, was therefore applied

15  to each point cloud to prevent recording erroneous pixels in the resulting rockfall inventory. Combined with a cell size of 0.15 m, this provided a minimum detectable rockfall across the survey area of $6.75 \times 10^{-4}$ m$^3$. More than 180 000 detachments were detected using the highest frequency of scans (~ hourly) over the 10-month monitoring period. The spatial and temporal distributions of rockfall observed are shown in Fig. 9.

In order to assess the influence of more frequent monitoring on the resultant volume frequency distribution, two inventories

20  were compared. These were analysed over the same monitoring duration, using scans separated by different intervals ($T_{Int}$) $T_{Int}$ < 1 h (hours) and $T_{Int}$ = 30 d (days). Figure 10a shows that an increase in the number of small rockfall and the proportional contribution of small events to the overall rockfall volume distribution is evident at $T_{Int}$ < 1 h. The power law scaling exponent,



$\beta$, increases from 1.78 (30 d) to 2.27 (< 1 h). Notably, while a rollover occurs at $T_{Int}$ = 30 d, this is not apparent at $T_{Int}$ < 1 h. Given that both sets of scans were processed using the same LoD for change detection, the comparison at this site demonstrates that the observed rollover must be due to superimposition and coalescence of events when longer survey intervals are used.

For all rockfall observed at $T_{Int}$ < 1 h, volume error was modelled according to Eq. 19 (Fig. 10b). Larger rockfall volumes exhibit a smaller error in proportion to their volume. Importantly however, as the vast majority of rockfall volumes are between 0.001 m$^3$ (a minimum of two pixels) and 0.01 m$^3$ (a minimum of 14 pixels), the uncertainty in volume ranges between 80% and 160% of the estimate. A consequence is that the total volumetric uncertainty over 10 months of the $T_{Int}$ < 1 h rockfall inventory is greater than that collected at $T_{Int}$ = 30 d. High frequency monitoring where change is dominated by a high frequency of low magnitude events is not well suited to accurate measurement of total change through time. The error estimates demonstrate that the uncertainty in volume is greatest for the datasets where $T_{Int}$ is low. For the highest frequency dataset, the total estimated volume is 110.87 ± 52.44 m$^3$ (± 47%), while the total estimated volume for the 30 d dataset is 72.37 ± 27.51 m$^3$ (± 38%) (Fig. 11). In summary, magnitude-frequency analysis of rockfall volumes (Fig. 10a) indicates that more frequent scanning detects a greater proportion of smaller rockfall events. Consequently, more frequent scanning also presents increased uncertainty in cumulative volume. Cumulatively, this error can be significant relative to the total over the monitoring period given that the size distribution of rockfall volumes adheres to a power-law.

## 5 Discussion

Improvements in near constant point cloud acquisition currently outstrip the development of standardised approaches to data treatment and analysis (Eitel et al., 2016; Kromer et al., 2017). Near constant scanning has the potential to generate a considerable number of point clouds (> 10$^3$-10$^4$), more than two orders of magnitude higher than many previous LiDAR monitoring campaigns (e.g. Teza et al., 2007; Abellán et al., 2010; Rosser et al., 2013; Royán et al., 2015). Key attributes of the techniques developed to process such datasets therefore relate to computational efficiency, the ability to automate processing, and minimising the accumulation of error between each survey pair. These have necessitated tailored approaches to LiDAR processing, which may differ from previous applications.

The relative gains of each processing step applied are evident in a gradual improvement to the applied LoD. A lowering of 30% with the application of radiometric and morphological filters occurs due to the removal of points considered to be both less accurate and less repeatable. The removal of these points prior to change detection, as opposed to the post-filtering of erroneous change measurements, was undertaken to minimise their impact upon normal estimation and 3D change detection. Their identification may also be used in alternative processing techniques; for example, as members of the fuzzy inference approach developed by Wheaton et al. (2010) to quantify spatially variable DoD uncertainty. The approach to change detection adopted in this study also yielded an improvement to the LoD, highlighting the importance of cylinder length for scenarios in which multiple surfaces may be intersected by the same normal vector in the resulting point cloud. Such surfaces can be





characterised as increasingly three-dimensional at the scale of the applied cylinder length (e.g. Brodu and Lague, 2013) or as rough surfaces that are surveyed at oblique viewing angles (Hodge et al., 2009). This problem is exacerbated for near constant monitoring, given that scanning from a fixed position increases the width of occluded zones on surfaces inclined away from the scanner, yielding higher offsets between measured surfaces inclined towards the scanner. Critically, adaptive change

detection techniques are necessary to account for the variability in point cloud quality across surfaces surveyed from a fixed position, where the installation location may yield unfavourable target geometries. These may take several forms, including varying cylinder lengths, widths, or spatially variable LoDs.

The applied method removes scans undertaken during rain/fog in order to preserve the accuracy of the resulting rockfall inventory. This limits the ability to relate rockfall occurrence derived from near-constant monitoring to real-time rainfall

datasets. Techniques that can operate during inclement weather conditions, such as ground-based InSAR, are therefore better suited to maintaining temporal consistency in rockfall datasets. While the precision and frequency of InSAR monitoring surpasses that of LiDAR instruments, highly precise change measurements are spatially averaged across large pixel sizes, resulting in a minimum detectable rockfall volume several orders of magnitude higher ($\sim 1 \ m^3$) as compared to terrestrial LiDAR datasets. While small magnitude changes that occur across large areas can be accurately characterised at high

frequency, detecting the frequency density of small erosion events is compromised. Point cloud generation for slope monitoring has been supplemented in recent years by the development of new photogrammetric techniques, in particular Structure from Motion (SfM; Niethammer et al., 2011; Westoby et al., 2012 Lucieer et al., 2013; Turner et al., 2015; Carrivick et al., 2016). When imagery is acquired from Unmanned Aerial Vehicles, SfM has the advantage of far lower operational costs than TLS, minimising areas of occlusion that occur from ground-level monitoring, and providing highly dense point clouds due to the

potentially small distances between the UAV and the slope. At present, however, the technique requires further development before it can be deployed for near constant monitoring of a surface.

Surface models exhibit a degree of uncertainty irrespective of the monitoring instrument and change detection technique used (Brasington et al., 2003; Lane et al., 2003; James et al., 2012). If analysis of time series data is required, error increases in proportion to the number of comparisons undertaken. For example, if the error between scan pairs is assumed to be constant,

monitoring over the same period using 1 000 scans produces 1 000 units of error, while monitoring over the same period using scans at the beginning and ending produces only two. At present, there is no consensus method to overcome this, even with the application of 4D smoothing techniques (Kromer et al., 2015). This suggests that monitoring at lower frequencies may provide more accurate estimates of rates of total change over longer periods. This is related to both the longer and hence time-averaged conditions captured, but also to the fact that the same level of change measured infrequently has less volumetric error

than when measured frequently, particularly when change is accrued by many small, discrete events. A decrease in $T_{Int}$ that approaches near-constant monitoring, 1 h, results in a shift in the exponent of the inverse power law of rockfall volumes from to 1.8 (30 d) to 2.3 (1 h). With a maximum plausible volume error, uncertainty in total rockfall volume ranged from 20% -

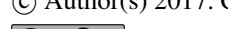



160% of the measured volume. Although critical to measure the full rockfall volume distribution, high frequency monitoring in this setting is not suitable for measuring net volume accumulated through large numbers of small events.

Given that small events present the highest aerial, and hence volumetric, uncertainty, the recent development of 3D volume estimation from point clouds (Carrea et al., 2012; Benjamin et al., 2016) may play an important role in near constant scanning.

The uncertainty of these techniques is determined by the precision of the point cloud, thereby eliminating uncertainty in object aerial extents due to linear interpolation into a fixed grid. However, these techniques also contain uncertainties, which arise in part from the meshing approach adopted (Soudarissanane et al., 2011; Hartzell et al., 2015; Telling et al., 2017). Due to the dependence of these techniques on a minimum of four points to create a closed hull, fully 3D techniques also are limited in their ability to resolve small, single point detachments. The development of scanners with increasingly small angular step

widths and increased rates of point acquisition, however, will decrease the minimum resolvable detachment. At present, the 3D clustering required to isolate points belonging to geomorphic change, combined with subsequent meshing of these points, comes at a considerable computational cost. These techniques therefore remain to be applied for > 10 scans (e.g. Carrea et al., 2012; Benjamin et al., 2016; van Veen et al., 2017).

This study demonstrates the need to adjust the frequency of data collection and processing according to the study aim. Here,

monitoring has been undertaken to detect near instantaneous discrete changes to the slope (rockfall) where both the spatial and temporal resolution of monitoring are important. Longer-term total change is more prone to error when change accrues from many small events, and big changes can occur in both the short and long term. There is a lack of research into this trade-off in spatial and temporal resolution but approaches that allow this to occur would be helpful in future. The collection of a high frequency time-series of scan data presents the opportunity to reduce uncertainty by averaging point positions through both

time and space as points are independent in neither space nor in time. This averaging can take the form of averaging the 3D position of each point, as utilised here, and in M3C2 (Lague et al., 2013), or the averaging of differences between points (Abellán et al., 2009; Kromer et al., 2015). Kromer et al. (2015) devised a method of averaging the distance between point clouds, whereby the change between them was assigned based on the median change for a neighbourhood of points along the normal direction.

While the precise magnitude-frequency exponent reported is specific to East Cliff, the scale-invariant behaviour of rockfall is similar to that observed in other rockfall inventories, albeit across a narrower range of magnitudes. Along the same stretch of coastline, previous monthly monitoring has yielded exponents of $\beta$ = 1.43-1.91 (Rosser et al. 2007; Barlow et al., 2012), similar to that identified in this study using the $T_{Int}$ = 30 d inventory ($\beta$ = 1.81). Both the exponent and presence of a rollover show a dependence upon monitoring interval at temporal scales considerably lower than the intervals used by Barlow et al. (2012).

From the derivatives of the magnitude-frequency distribution, it is therefore apparent that risk increases by $10^3$ if calculated based upon event frequency, when the rock face is monitored more frequently. For rockfall distributions created from TLS surveys, Young et al. (2011) noted that the ability to resolve small-scale changes should not introduce a rollover, because the





smallest reported rockfall is larger than the minimum detectable event identifiable in change mapping. From a statistical perspective, this statement holds true as long as the frequency density is not estimated using a moving kernel, which enforces an extrapolation of density that extends one kernel half width beyond the range of the observations both below the minimum and above the maximum, introducing inflections in the frequency density at the tails (Lim et al., 2010). Here, a rollover in the

magnitude-frequency distribution is identified for the $T_{Int} = 30$ d inventory. However, this rollover was not present in the $T_{Int} < 1$ h dataset. Only rockfall larger than the 0.03 m (LoD) $\times$ 0.15 m $\times$ 0.15 m (the area of each cell, which exceeds the minimum point spacing) were analysed, which equates to a volume of $6.75 \times 10^{-4}$ m$^3$. Critically, this implies that where events coincide, or coalesce, (here $< \sim 0.01$ m$^3$) or where the mechanisms driving change are not spatially independent, event frequency is partially determined by survey interval. Given the development of spatially contiguous rockfall scars that has

been observed in this setting (Fig. 9) and in other studies (Rosser et al., 2007; 2013; Stock et al., 2011; Kromer et al., 2015; Rohmer and Dewez, 2015; Royán et al., 2015), the creation of magnitude-frequency distributions from near constant monitoring has the potential to generate improved understanding of the mechanisms of geomorphic change.

## 6 Conclusion

The magnitude-frequency distribution of geomorphic change is an important descriptor of the relative efficacy of event sizes,

and the nature of the hazard that they pose. Improvements in the ability to resolve the magnitude of events have surpassed those relating to our ability to constrain event frequency. More frequent monitoring of a changing surface increases the cumulative error over comparable monitoring periods. This study has reduced this error through a workflow that filters features based on their reliability, or repeatability between point clouds. The use of a morphological and radiometric filter identifies such points that belong to such features, in particular edges, surfaces of high incline and vegetation. Surfaces that appear three-

dimensional at the scale of the cylinder length used in 3D change detection are also accounted for, by drawing upon a variable length cylinder. This avoids uncertainty in change detection between point clouds of an unchanged surface, in which multiple surfaces may be intersected by the same normal vector. Having been applied to an actively failing rock slope, this adaptation of the M3C2 algorithm (Lague et al., 2013) yielded a fivefold decrease in uncertainty between point clouds. The procedure was applied to a dataset of $10^3$ point clouds. Without significant morphological change to the monitored surface, alignment of

this number of scans was undertaken to the first scan of the dataset given that this avoids drift due to atmospheric variation but does not detriment the alignment accuracy. Rockfall inventories collected at $T_{Int} < 1$ h and $T_{Int} = 30$ d highlight that more frequent monitoring discerns a higher proportion of small rockfall than at $T_{Int} = 30$ d. Both the size and shape of rasterised events determines the ability to accurately quantify their volume, with smaller events and events with higher perimeter to area ratios yielding a higher degree of uncertainty. A means of quantifying this uncertainty has therefore been proposed, which is

necessary to carry forward into any quantification of volumetric loss during near constant monitoring campaigns.



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



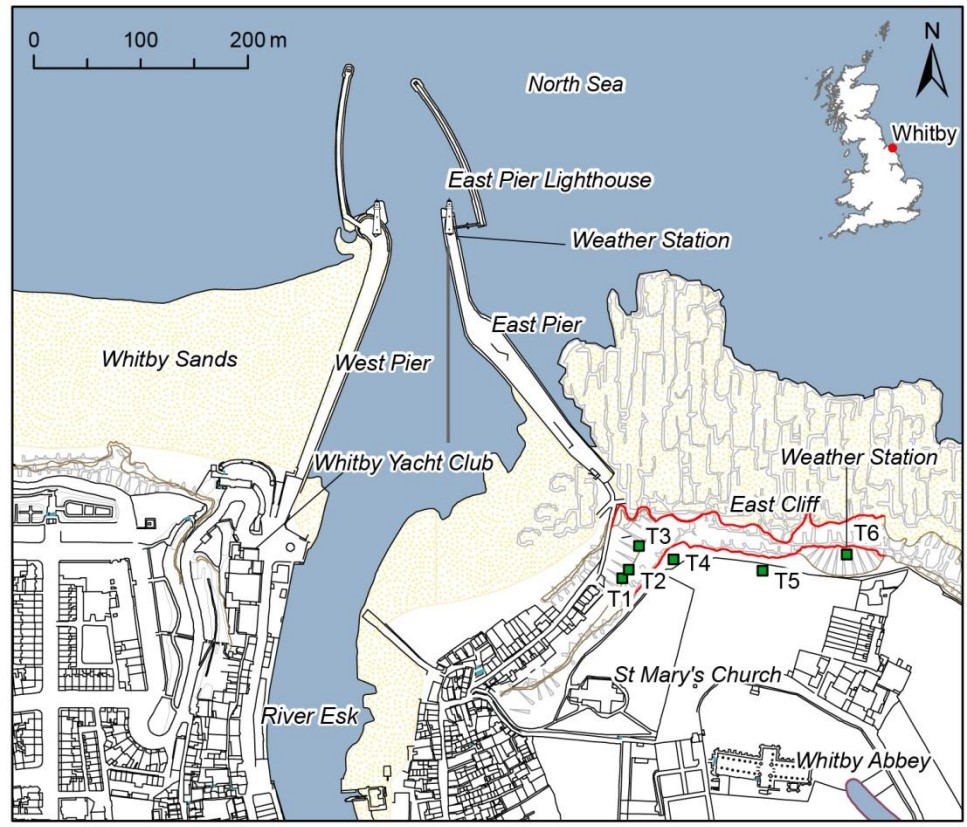

**Figure 1: Map of Whitby with the area scanned delineated with red lines. The Riegl VZ-1000 scanner is installed within East Pier Lighthouse, located at the end of East Pier. The targets installed for the SiteMonitor4D system, in addition to the weather stations, are illustrated (T1 – T6). Whitby Abbey lies 180 m from the cliff top. Weather stations, powered using the solar panels, are located on East Cliff and the lighthouse. The fuel cells were used to power the webcam, tablet and scanner and Wi-Max connection; however, this is now powered by mains supply. The tablet runs SiteMonitor (3D Laser Mapping Ltd.), and saves the scan data to a Dropbox™ folder for upload through the yacht-club broadband. Summary statistics of the scanning schedule along with the weather station data are uploaded to GeoServer using the broadband connection, where it can be accessed through GeoExplorer (NavStar Geomatics). Map produced using shapefiles from Ordnance Survey © Crown Copyright and Database Right 2016. Ordnance Survey (Digimap Licence). System workflow modified from an original design by NavStar Geomatics.**



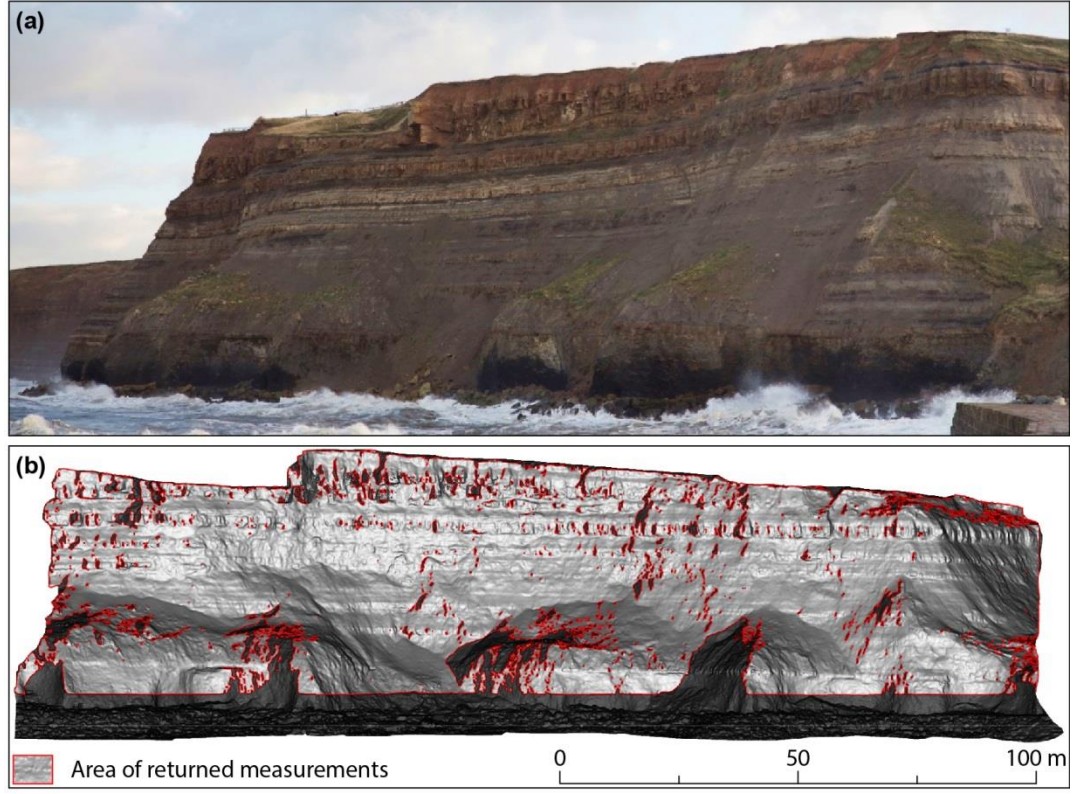

Area of returned measurements

0          50          100 m

**Figure 2: (a) Image of the cliff taken 1 h before high tide on 25h November 2015. Horizontally bedded strata are evident, with upper beds stained orange from downslope wash from glacial till of variable depth. The lower buttress comprises shales and some sandstone, while the near-vertical upper portion of the cliff comprises outcropping sandstone, and sandstone interbedded with carbonaceous muds. (b) Area scanned by the permanent monitoring system used in this study (light grey) draped over a complete slope model of the entire cliff captured from multiple positions along the foreshore (dark grey). The total area measured is 8 561 m$^2$, 89% of the cliff face total (9 592 m$^2$). Cliff is ~ 210 m across and ~ 60 m high.**



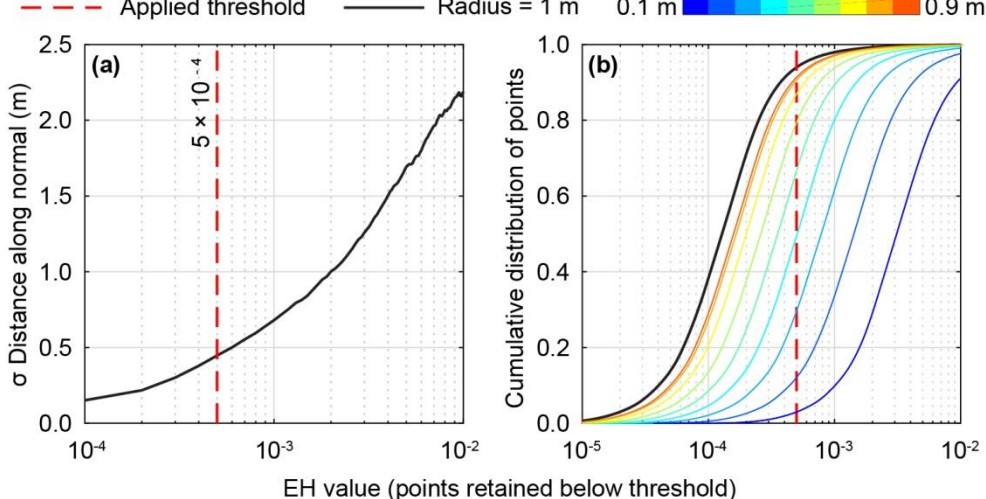

**Figure 3: (a) A change detection is undertaken between two point clouds where no observable movement occurred. The standard deviation for a single point cloud therefore indicates the level of noise between the two. This value is estimated by first including only points with the lowest edge/hole values (EH 10-4), and then including points with increasingly large edge/hole values (up to EH 10-2). When high EH values are retained, the total error, defined by the standard deviation of change, increases. Edge and hole values for the point cloud are retained below the position of the threshold (red dashed line). (b) The cumulative proportion of EH values within an entire point cloud. Each line represents a different neighbourhood radius search. For the same points, EH values are lower using a larger search radius because more neighbouring points (k in Equation 1) are found. An inflection in the number of points retained is used to define the threshold at 95%. This ensures that artefacts such as holes are not introduced into the point cloud by removing too many points. While the *EH* values change, their distribution across the point cloud remains the same due to the normalisation by point density. A 1 m radius was selected to ensure that a minimum of four points, the minimum needed to estimate the *CoG* in addition to the query point, would always be found.**



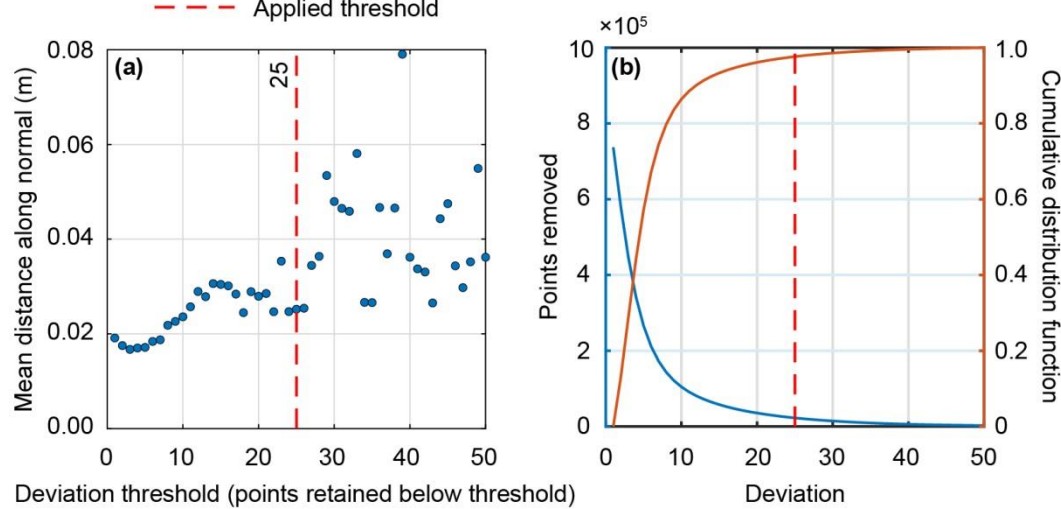

**Figure 4: (a) Mean absolute distance between two point clouds with no observable change. Similar to Fig. 3, this indicates the comparison uncertainty between both scans. The mean distance is calculated for change estimates attributed to points with each deviation, from 1 – 50. Error increases from** ca. **0.03 – 0.06 m at values > 25. The variability in error also increases such that the selection of an appropriate threshold > 25 is not possible. (b) The number of points removed (blue) alongside the cumulative distribution of deviation values. A threshold of 25 ensures that only 2% of points are removed.**

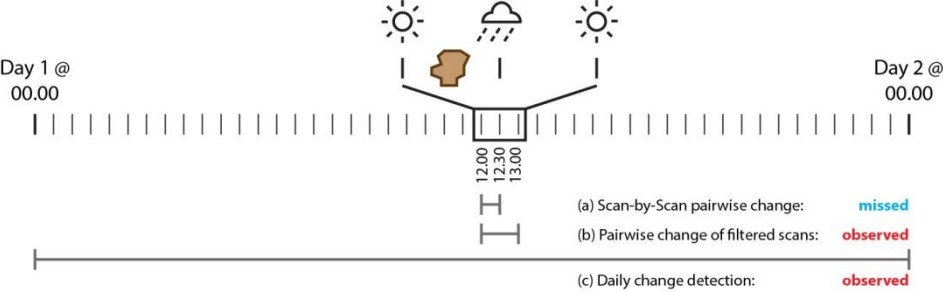

**Figure 5: Conceptual illustration of the significance of removing partial scans. While parts of these scans provide accurate estimates of surface change, if a rockfall occurs in an area of no data, the failure will be missed using pairwise change. These scans must therefore be removed prior to change detection of the scan database.**



**Figure 6: (a) The radius for each point on the cliff at which the point clouds is most planar, with a mean value of 1.1 m, used to estimate the normal vector prior to change detection. This point cloud was used as a reference model, such that the normal radius of points in subsequent scans was assigned based on the radius of closest point in this scan, (b) Surface planarity at a radius of 1 m, where higher values indicate a more 3D neighbourhood. These occur at inflections in slope profile and in areas of high local relief, such as the sandstone beds near the cliff-top. Gaps in point cloud are zones of occlusion illustrated in Fig. 2b.**





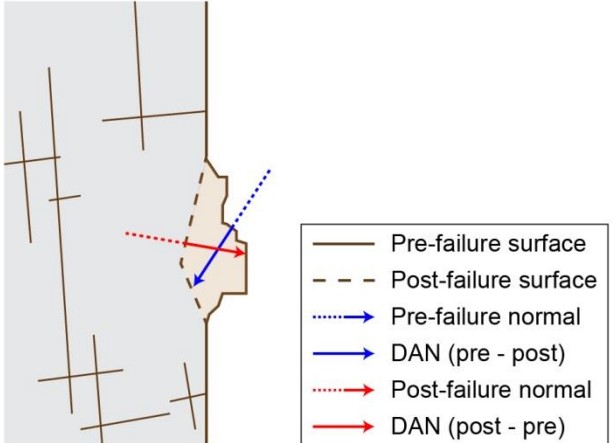

**Figure 7: Conceptual variation in the distance along the normal for a rockfall. The normal direction estimated using a planar, post-failure surface more accurately represents the direction of change than the post-failure surface vector, due to the complexity of the pre-failure surface. The difference in vector lengths also illustrates the sensitivity of the 3D change measurements to the normal estimation. Dashed lines indicate that normal directions are subject to sign ambiguity. The cylinders used in the change detection (next section) are represented by the solid arrow.**



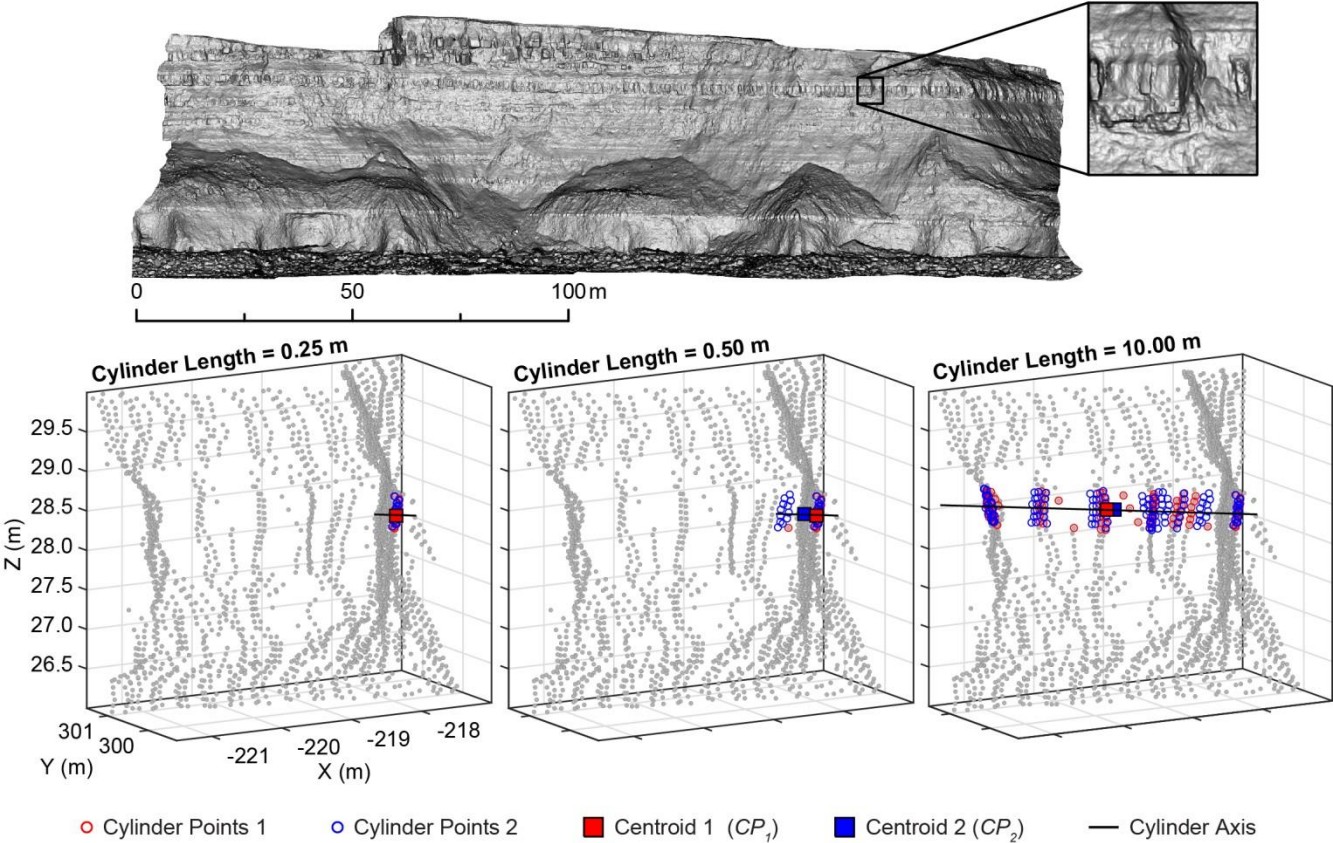

**Figure 8: Inputs used for distance estimation with varying cylinder lengths. No appreciable change occurred between these two scans. As the cylinder length increases (from 0.25 m to 0.50 m to 10 m), the number of surfaces that the cylinder intersects increases (direction equal to the normal vector). All points within a 0.25 m radius would be included as cylinder points (circles), and the distance between their mean positions (squares) calculated. From top to bottom, this distance is 0.0011 m to -0.1460 m to 0.0938 m. Longer cylinders intersect multiple surfaces and therefore measure the distance between projected centroids that do not accurately represent the surface to which the query point belongs.**



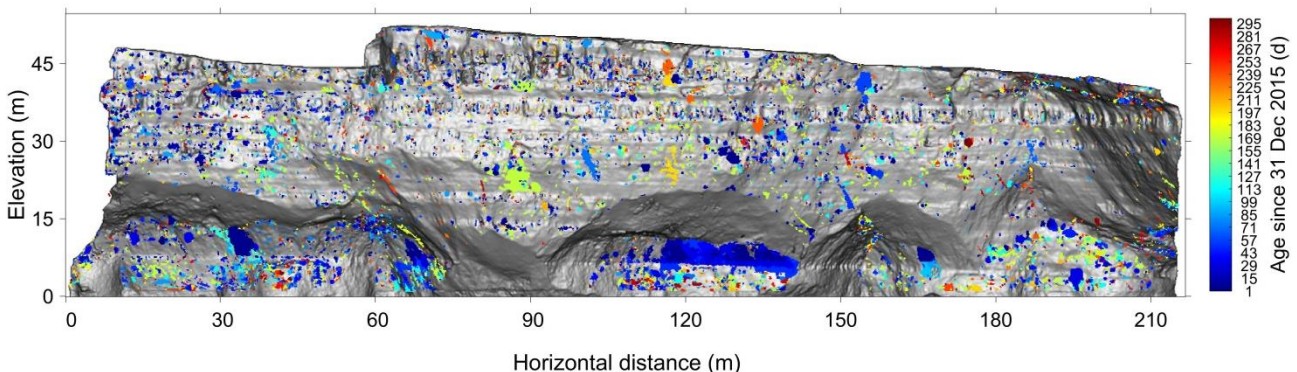

**Figure 9: Distribution of rockfall across East Cliff monitored at sub-hourly intervals between 5th March 2015 and 30th December 2015. Rockfall are distributed across the entire cliff face, in particular in areas of exposed bedrock. Although the high water mark is below the portion of cliff shown in this Fig., the largest and most frequent rockfall occur at the base of the cliff. Accumulation and loss of material in the areas of non-exposed bedrock on the cliff buttresses, which runs across the cliff face at ca. 17 m elevation, were removed. Colours represent the age since 31st December, where red represents the oldest rockfall.**

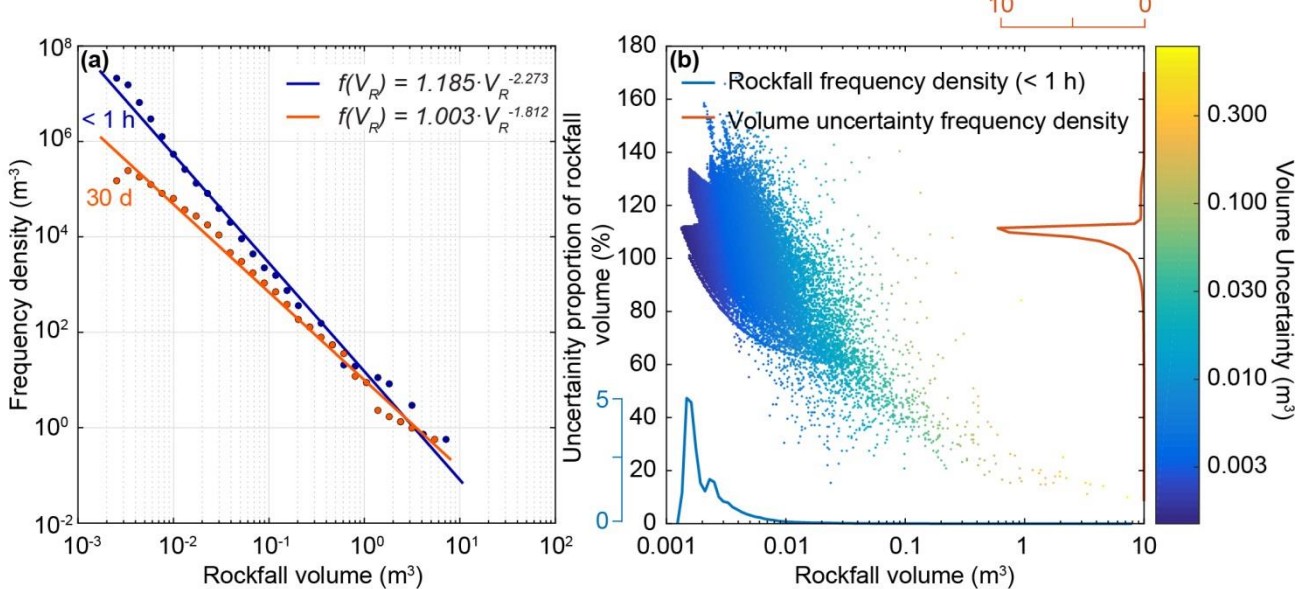

**Figure 10: (a) Magnitude-frequency distribution for rockfall inventories acquired at varying $T_{Int}$ showing that a higher proportion of small events is established by monitoring at high frequencies. Rockfall used range from 5th March 2015 – 30th November 2015 to enable direct comparison between $T_{Int} < 1$ h and nine $T_{Int} = 30$ d change detections. (b) Rockfall volumes from < 1 h rockfall inventory. Percentage volume error is estimated using the LoD, number of internal pixels and number of edge pixels. Frequency densities (kernel density estimates) are appended to each axis, showing that rockfall volumes < 0.01 m³ account for the greatest proportion of measured rockfall (modal volume = 0.0081 m³). As a result, errors range from ca. 60% to ca. 140% for most rockfall (modal error percentage = 109%). Cumulative volume estimations using rockfall of this size may vary by at least the actual volume.**





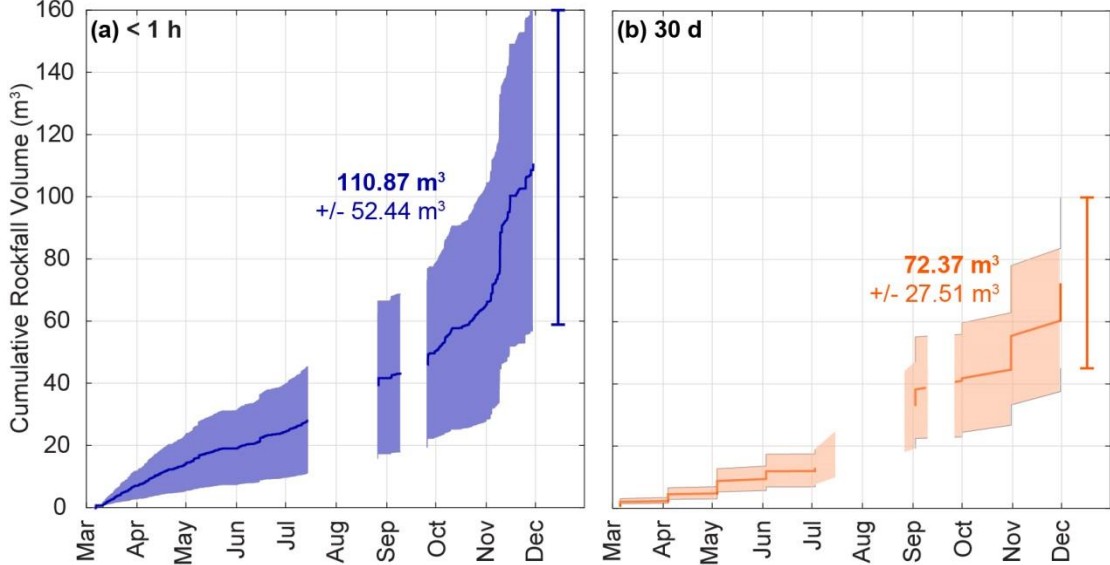

**Figure 11: Cumulative rockfall volumes measured though the monitoring period, using data from all 11 monitoring intervals. The results show that far higher volumes of material, up to twice those recorded by 30 d monitoring, are measured at sub-daily intervals. The times of pairwise change detections are recorded as the date of the first scan, rather than the second. As a result, although all scan intervals record a significantly increased rate of rockfall activity during November; this appears earlier on the plot for longer scan intervals. The total estimated volumes are not included for comparison as change detections cannot be recorded up to the final day of monitoring for longer time intervals (30th December).**



**Acknowledgements**

This research formed part of a Ph.D. studentship provided by the Department of Geography, Durham University, and ran alongside the Knowledge Transfer Partnership (KTP8878) awarded to NJR, RJH, AA and 3D Laser Mapping Ltd. We thank ICL Fertilizers (UK) Ltd for ongoing support of this project. Assistance in the development of this system was provided by

5   Navstar Geomatics, and the maintenance and running of the system was supported by S Waugh and D Hodgson. Data is available on request from the authors.