# Peer review of "Optimising 4D Approaches to Surface Change Detection: Improving Understanding of Rockfall Magnitude-Frequency"

_Earth Surface Dynamics, 2017_

## Referee Comment (RC1) · M.-H. Derron (Referee) · 8 Sep 2017

Two 2017 papers bring a new dimension to the research in rock slope processes thanks to near-continuous TLS monitoring. The first one is from Kromer et al. (also in Earth Surf Dyn), and the second one is the present contribution by Williams et al.

In this contribution, authors have acquired a unique dataset of terrestrial LiDAR scans (almost 9000 scans of a coastal cliff, with 1 hour interval over 10 months). The present paper is mostly dedicated to the methods that had to be developed to process such an exceptional dataset (filtering, alignment, change detection and volumes estimation). However, from a geomorphological point of view, this contribution already sheds a new

light on an old problem: the rollover effect in frequency-magnitude distributions. In my opinion, main original contributions of this paper are: - Filtering of edges and of bad quality returned pulses before alignment of point clouds - An improved M3C2 algorithm for change detection - Original results on the influence of sampling frequency on (a) frequency-magnitude distributions and (b) cliff retreat rates estimations.

In my view the main limitation of the method proposed here is to pass by a raster to estimate volumes. However this point is properly discussed, and an error on the volume estimation is proposed. Considering the high quality of this contribution, I only have some minor corrections to suggest: - I had to read several times the text of page 5 and the caption of Fig3 before to understand the meaning of this figure. It would be useful to develop a bit more this part. I find it confusing in the present form. - Fig 7: I did not get the complete message of this caption. May be you can split the different elements in several sub-figures (pre vs post, sign ambiguity, sensitivity). Acronym DAN not defined (dist along normal) - Part 3.4 – M3C2: it would help to have an extra figure here to explain the geometry of the cylinder and its relations with the point clouds - Acronyms LoD and CoG are not defined at their first occurrence. - P13, L 23: "N is the number of cells that delimit the event" ; Isn't it a bit ambiguous to use the verb "delimit" to define N, with regards to Nb ? - Complete Eq 16 and 18c with the results of the calculations - P14, L14: I would "remove the areas BELOW the error threshold" (?) - P18 L30: the term "risk" is not appropriate here as it includes many other things than the block release frequency. Suppress this sentence.

---

## Author Comment (AC1) · 13 Sep 2017

We thank the reviewer for his constructive comments. Responses to the comments and suggestions made in Paragraph 3 of the review are provided below, along with the proposed amendments (italicised).

"In my view the main limitation of the method proposed here is to pass by a raster to estimate volumes. However this point is properly discussed, and an error on the volume estimation is proposed".

We agree that this presents a limitation and have highlighted the potential for 3D volume estimation, with reference to recently published work on this topic. As discussed in the text, rasterising was undertaken due to the structure (primarily occlusion) and high frequency of data, which necessitated an efficient processing algorithm.

"I had to read several times the text of page 5 and the caption of Fig 3 before to understand the meaning of this figure. It would be useful to develop a bit more this part. I find it confusing in the present form."

The reviewer highlights some confusion regarding Fig. 3 (the selection of a threshold edge-hole value) based on its description in text and in the figure caption, specifically around the function of this step in our processing. In light of this, in text amendments are provided below. Additionally, we feel that examples of East Cliff coloured by the EH value would be ideally suited to a supplementary information, alongside the same figure coloured by the deviation filter. Both figures are also provided below.

Text to add - P13 L20: In order to select a threshold EH value, the epistemic error between two point clouds was quantified once EH values above a specific threshold were removed. As the threshold is increased, fewer points are filtered from each cloud. The theoretical distance between the two point clouds is assumed to be zero given that no rockfall were observed between their collection. Any offset therefore represents the epistemic uncertainty, which is quantified for this purpose as the standard deviation of a 3D change detection. This uncertainty is plotted against the applied threshold in Fig. 3a. As the threshold is increased, points with higher EH values are retained and the offset between the two point clouds also increases. The distribution of EH values across the cloud is presented in Fig. 3b. Using a 1 m search radius, an inflection at the 95th percentile of points occurs at  $5 \times 10^{-4}$ . As a threshold, this value typically removes 5% of points, which, as depicted by the dashed line in Fig. 3a, account for uncertainties > 0.5 m. In addition to identifying edge features on the cliff face, this also helps to delineate areas of occlusion within the point cloud. The point density in Eq. 1, k, is used to filter spurious 'floating points' in the dataset (for example birds or dust). k values

---

## Referee Comment (RC2) · 10 Oct 2017

The paper entitled "Optimising 4D approaches to Surface change detection: improving understanding of rockfall maginute-frecuency" by Williams et al. presents an interesting contribution about a near-continuous monitoring of a cliff using point clouds captured by means of a TLS. The paper addresses the well-known relationships between magnitude-frequency in geomorphological processes but using a very valuable dataset of scans acquired every hour for a period of several months. New technologies are now providing this kind of huge datasets and the key questions in the upcoming years would be relate to the best practices to manage large datasets. In this line, some ideas and

procedures are presented in the paper (e.g. filtering the cloud previous to change detection). I recommend the publication of the paper after some minor changes. In this line, I suggest improving some figures (please see comments below) although in general figures are of high-quality (e.g. figure 9 is great). I also recommend a clear statement of the objectives of the paper at the end of the first section. The methods section is clear but I think it could be improved using a workflow chart summarizing the different steps in one figure. I think this would be really helpful for readers. Finally, I would recommend a deeper discussion about the effect of using a spatially homogeneous LoD for change estimation in such a heterogeneous surface (including different layers-lithology with different slopes, small local faults, etc.). Detailed comments or suggestions:

INTRODUCTION: I miss here a clear statement of the objectives of the paper. I know that at the end of the section (i.e. lines 28-30) there is a description about what is done in this work however it sounds to me more introductory to a methodological section than presenting the main goals of the paper. I recommend to include a paragraph with a clear statement of the objectives. The classical reader will expect this at the end of an introduction. Fig.1: can be improved. England is floating in the same scale-map? Please use a box to delimitate England in a location box. The rest of the map needs a legend. Provide a legend indicating the meaning of symbols presented (green dots, red lines, etc.). The caption is huge and some information is not necessary, e.g. "powered by solar panels" Fig. 2b I recommend saying "hillshade of the cliff showing the area covered by the TLS" in the caption. Fig. 3 Caption: please use the superscript instead 10-4. Please indicate in the graph that colors in the figure 3b represent different search radius. Page 12, L20: the acronyms LoD and DoD show up before the complete term is presented (the term is presented in section 3.5, page 13 which later, please check the use of acronyms along the text) Section 3.5: do you assume a homogeneous LoD? This argument deserves a deep discussion. Section 4: the first sentences of the first paragraph sounds again like methods. The real results section starts in line 16.

**ESurfD**

---

## Author Comment (AC2) · 21 Oct 2017

**Williams, J. G., Rosser, N. J., Hardy, R. J., Brain, M. J., and Afana, A. A.: Optimising 4D Approaches to Surface Change Detection: Improving Understanding of Rockfall Magnitude-Frequency, Earth Surf. Dynam. Discuss., https://doi.org/10.5194/esurf-2017-43, in review, 2017.**

**Response to Reviewer 2.**

We thank Reviewer 2 for the constructive and positive review of our manuscript. Below we reproduce the reviewer comments (*italics*) followed by our responses, illustrated with the amendments that we intend to make to the manuscript as a result (in quotation marks " … ").
* * *
- *INTRODUCTION: I miss here a clear statement of the objectives of the paper. I know that at the end of the section (i.e. lines 28-30) there is a description about what is done in this work however it sounds to me more introductory to a methodological section than presenting the main goals of the paper. I recommend to include a paragraph with a clear statement of the objectives. The classical reader will expect this at the end of an introduction.*

To clarify the aim of our paper, we will add the following statement at the end of the Introduction:

P3 L29: "This paper presents a technique for change detection from near-continuously collected 3D point cloud data. The dataset includes ~ $10^3$ individual point clouds, with each comprising > $10^6$ points. Using this method, we demonstrate the influence of survey frequency on firstly the magnitude frequency of rockfall, and secondly on the uncertainty associated with measuring volumetric change through time."

- *The methods section is clear but I think it could be improved using a workflow chart summarizing the different steps in one.*

A workflow diagram will be added, as below. Figure numbers will be amended accordingly.

[Figure]

Scan acquisition and data transfer

↓

*.RXP to *.MAT conversion
(including RCF application)

↓

Rotation

↓

Edge-Hole filter

↓

Deviation filter

↓

ICP alignment

↓

Octree creation

↓

Normal estimation

↓

3D change detection

↓

Change raster creation

↓

LoD binary creation and
rockfall vectorisation

Figure 3: Flow diagram representing the stages of rockfall inventory compilation. All stages following ASCII to MAT conversion were written in MATLAB, with ICP alignment and rockfall vectorisation using the built-in functions *pcregrigid* and *bwboundaries*. Point clouds are initially rotated to become approximately planar across the *x-z* plane, enabling the removal of points outside a tight bounding cuboid and rasterising of the point clouds of change. This rotation also enables an efficient solution to subsequent normal direction ambiguity, where the *y* component of each normal vector should always point out of the surface.

- *Fig.1: can be improved. England is floating in the same scale-map? Please use a box to delimitate England in a location box. The rest of the map needs a legend. Provide a legend indicating the meaning of symbols presented (green dots, red lines, etc.). The caption is huge and some information is not necessary, e.g. "powered by solar panels"*

A revised version of original Figure 1 is provided below, which includes delimiting the inset map of the UK, addition of a legend, and the removal of superfluous detail, as suggested by the reviewer. A shortened caption is also provided.

[Figure]

Figure 1: Map of Whitby with the area scanned delineated with red lines. A Riegl VZ-1000 scanner is installed within East Pier Lighthouse. The targets installed for the SiteMonitor4D Range Correction Factor estimation are illustrated (T1 – T6) in addition to the weather stations. Whitby Abbey lies 180 m from the cliff top. Map produced using shapefiles from Ordnance Survey © Crown Copyright and Database Right 2016. Ordnance Survey (Digimap Licence).

- *Fig. 2b I recommend saying "hillshade of the cliff showing the area covered by the TLS" in the caption.*

We will modify the caption to Figure 2(b) as follows: "(b) Slope model of the cliff showing the area covered by the TLS (light grey) draped over a 3D model of the cliff, surveyed from multiple positions along the foreshore (dark grey). The total area measured is 8 561 $m^2$, or 89% of the cliff face area (9 592 $m^2$). The cliff is ~ 210 m across and ~ 60 m high."

- *Fig. 3 Caption: please use the superscript instead 10-4. Please indicate in the graph that colors in the figure 3b represent different search radius.*

This error appears to have arisen during conversion to a *.PDF. The superscript has been added in light of this comment. The figure has been amended to more clearly explain the variation in colours:

[Figure]

- *Page 12, L20: the acronyms LoD and DoD show up before the complete term is presented (the term is presented in section 3.5, page 13 which later, please check the use of acronyms along the text)*

This has also been identified by Reviewer 1. The following amendments have been made:

P6 L28: "… thereby lowering the Level of Detection (LoD) that could be applied …"

P5 L5: "… central position of the neighbourhood points (CoG) calculated …"

- *Finally, I would recommend a deeper discussion about the effect of using a spatially homogeneous LoD for change estimation in such a heterogeneous surface (including different layers-lithology with different slopes, small local faults, etc.). Section 3.5: do you assume a homogeneous LoD? This argument deserves a deep discussion.*

We agree that this requires a more full discussion. We add the following at the point in the manuscript when the Level of Detection (the LoD) is presented in the review of other approaches to change detection (Section 3.5), at P13 L9: "The delineation of areas of geomorphic change, here rockfall, involves masking regions of change that exceed a hard threshold at the level of detection (LoD), that is either estimated locally (e.g. Wheaton et al., 2010; Lague et al., 2013) or is estimated across the entire point cloud (e.g. Abellán et al., 2009). Methods that estimate spatially-variable LoDs have enhanced the ability to identify volumetric loss as compared to the application of a single LoD, with the latter set to exceed a significant portion of the modelled uncertainty across the area of interest. Across a rock slope, the likelihood of generating similar point distributions between surveys, which determines the accuracy of change detection, is primarily influenced by the target geometry relative to instrument position and surface reflectance characteristics. These properties may vary with lithology, the surface complexity and moisture, and survey geometry (Clark and Robson, 2004; Bae et al., 2005; Litchi et al., 2007; Kaasalainen et al., 2008; 2010; Pesci et al., 2008; 2011; Soudarissanane et al., 2011; 2016). These factors raise the potential for real change to be masked when using a single LoD but, equally, the application of a single LoD becomes increasingly computationally efficient when dealing with a large number of surveys. The benefits of using a single LoD are primarily in the consistency in measurement across the area of interest. For example, if the purpose of monitoring is to generate a rockfall inventory where

the relative magnitude of events is important, a single LoD ensures consistency in the minimum detectable rockfall across the area of interest and minimises the potential for recording erroneous events, which we demonstrate here will accumulate with an increasing number of surveys. A single LoD was therefore identified between scan pairs in which no rockfall occurred as two standard deviations of the 3D change, after Abellán et al. (2009). This was of comparable magnitude to the LoD recorded for every scan pair in the dataset; hence, the maximum-recorded LoD was applied to all scan pairs in the dataset. Similar to Kromer et al. (2017), these change estimates are assumed to include the registration error, which is reduced here through range correction using fine-scanned targets and through ICP. For sites whose geometry creates a highly variable point spacing within a single survey, a spatially variable LoD is appropriate even for the purpose of compiling an inventory of geomorphic events, so long as a record of the LoD across the surface is kept. Open-pit highwalls, for example, typically comprise a series of benches to prevent rockfall from travelling further downslope. This design generates considerable variation in instrument-object distances across the slope and a spatially-variable LoD. More broadly, spatially variable LoDs can be considered better suited to measuring total erosion budgets across a single surface than the relative contribution of individual events of varying sizes."

- *Section 4: the first sentences of the first paragraph sounds again like methods. The real results section starts in line 16.*

The sentences (P15 L11-16) will be removed from Section 4 and will be inserted into section 3.5: "The pairwise change detection method described above was applied to a near-continuous monitoring dataset collected at East Cliff, Whitby. In total, 8 987 point clouds were collected and processed to generate an inventory of 3D rockfall geometries. The LoD was derived for every sequential scan to ensure that no increase in registration or epistemic errors developed through the monitoring period. This value lay consistently between 0.01 – 0.03 m. The maximum LoD, 0.03 m, was therefore applied to each point cloud to prevent recording erroneous pixels in the resulting rockfall inventory. Combined with a cell size of 0.15 m, this provided a minimum detectable rockfall across the survey area of $6.75 \times 10^{-4}$ $m^3$. More than 180 000 detachments were detected using the highest frequency of scans (~ hourly) over the 10-month monitoring period. The spatial and temporal distributions of rockfall observed are shown in Fig. 9.

In order to assess the influence of more frequent monitoring on the resultant volume frequency distribution, two inventories were compared. These were analysed over the same monitoring duration, using scans separated by different intervals ($T_{Int}$) $T_{Int} < 1$ h (hours) and $T_{Int} = 30$ d (days)."

**References used in this response**

**Abellán, A., Jaboyedoff, M., Oppikofer, T. and Vilaplana, J.M., 2009.** Detection of millimetric deformation using a terrestrial laser scanner: experiment and application to a rockfall event. Natural Hazards and Earth System Sciences, 9(2), pp. 365-372.

**Bae, K.H., Belton, D. and Lichti, D., 2005.** A framework for position uncertainty of unorganised three-dimensional point clouds from near-monostatic laser scanners using covariance analysis. International Archives of Photogrammetry and Remote Sensing, 36(3), pp. 7-12.

**Clark, J. and Robson, S., 2004.** Accuracy of measurements made with a Cyrax 2500 laser scanner against surfaces of known colour. Survey Review, 37(294), pp. 626-638.

**Kaasalainen S., Kukko A., Lindroos T., Litkey P., Kaartinen H., Hyyppa J., Ahokas E., 2008.** Brightness measurements and calibration with airborne and terrestrial laser scanners. IEEE Transactions on Geoscience and Remote Sensing, 46, pp. 528–534.

**Kaasalainen, S., Niittymaki, H., Krooks, A., Koch, K., Kaartinen, H., Vain, A. and Hyyppa, H., 2010.** Effect of target moisture on laser scanner intensity. IEEE Transactions on Geoscience and Remote Sensing, 48(4), pp. 2128-2136.

**Kromer, R.A., Abellán, A., Hutchinson, D.J., Lato, M., Chanut, M.-A., Dubois, L. and Jaboyedoff, M., 2017.** Automated Terrestrial Laser Scanning with Near Real-Time Change Detection - Monitoring of the Séchilienne Landslide. Earth Surface Dynamics, doi:10.5194/esurf-2017-6.

**Lichti, D.D., 2007.** Error modelling, calibration and analysis of an AM–CW terrestrial laser scanner system. ISPRS Journal of Photogrammetry and Remote Sensing, 61(5), pp. 307-324.

**Pesci, A., Teza, G. and Ventura, G., 2008.** Remote sensing of volcanic terrains by terrestrial laser scanner: preliminary reflectance and RGB implications for studying Vesuvius crater (Italy). Annals of Geophysics, 51(4), pp. 633-653.

**Pesci, A., Teza, G. and Bonali, E., 2011.** Terrestrial laser scanner resolution: numerical simulations and experiments on spatial sampling optimization. Remote Sensing, 3(1), pp. 167-184.

**Soudarissanane, S., Lindenbergh, R., Menenti, M. and Teunissen, P.J.G., 2009.** Incidence angle influence on the quality of terrestrial laser scanning points. In Proceedings of the ISPRS Workshop on Laser Scanning. Paris, France, 1 – 2 September. 38(W8), pp. 183-188.

**Soudarissanane, S., Lindenbergh, R., Menenti, M. and Teunissen, P., 2011.** Scanning geometry: Influencing factor on the quality of terrestrial laser scanning points. ISPRS Journal of Photogrammetry and Remote Sensing, 66(4), pp. 389-399.

**Wheaton, J.M., Brasington, J., Darby, S.E. and Sear, D.A., 2010.** Accounting for uncertainty in DEMs from repeat topographic surveys: improved sediment budgets. Earth Surface Processes and Landforms, 35(2), pp. 136-156.

---

## Author Response (AR1)

Dear Prof. Castillo,

Please find attached the revised manuscript, supplementary information, and abstract for our paper. The changes that have been made relate directly to those stated in our responses to the reviewers, which we hope are deemed sufficient. Should you require a Word document with these changes tracked and identified, please do not hesitate to contact us.

Many thanks,

Dr Jack Williams

---

## Author Response (AR2)

**esurf-2017-43**

Dear Dr Castillo,

We are grateful for your review of our manuscript, consisting of both general recommendations and detailed in text amendments provided in the supplement. Given the extent to which the manuscript has been restructured, we provide details of our amendments below as opposed to a document with tracked changes. These more closely align the manuscript to the I-M-R-D-C approach referred to in your review.

- **Title:** We have opted to amend this to *'Optimising 4D Surface Change Detection: An Approach for Capturing Rockfall Magnitude-Frequency'*. We feel that this better reflects the proportion of methodological analysis in the paper, without overly focussing on the interpretation of magnitude-frequency.

- **Abstract:** Remains unchanged.

- **Introduction:** As suggested, we have split the introduction into sections. This begins with an introduction to the size distribution of geomorphic events followed by the importance of the temporal resolution of monitoring in capturing this. While our introduction now contains a more even balance between M-F and the method, we feel that these initial sections are critical in underpinning our motive for near-continuous monitoring. We then proceed to describe the sources of uncertainty involved in near-continuous monitoring, which stem from scanning from a fixed position. As you suggest in your in-text comments, this section should come early in the paper to outline the types of uncertainty that we are attempting to minimise. Our final section outlines the uncertainty in volume estimation. At the end of each section, we describe an associated objective. These are:

  o To capture the influence of near-constant monitoring on the magnitude-frequency distribution of rockfall from an actively failing rock face.

  o To minimise the errors that arise from near-continuous monitoring, in order to minimise the minimum detectable movement.

  o To describe the impact that a changing magnitude-frequency distribution has on the overall uncertainty of eroded volume through time

- **Study site and data collection:** Unchanged

- **Method: Optimising event extraction from near-continuously collected point clouds**.

  o Here we have maintained the same sections but have subdivided Section 3.1 into the separate filters suggested (AOI extraction > edge-hole filter > waveform deviation filter > partially obscured point clouds. At the beginning of the section we have detailed with greater clarity the aim of the method and the chronology of the workflow.

  o At the beginning of each section, we have outlined the objective of the particular step and, where appropriate, previous research.

  o We have also sought to better highlight work that is novel and work that is not. As well as in text, novel inputs are summarised in Fig. 3, which is discussed in further detail later in this letter. In text, an example of this is for the waveform deviation filter *'While the sensitivity of the waveform to target geometry has previously been highlighted (Williams et al., 2013), it has not previously been documented as a method to filter points'*. Importantly, where we refer to near-continuously collected data or to large numbers of scans, the method described is indeed newly presented in this manuscript. For example, we create a reference map of the optimal search radiuses across every point cloud. Such an action is not necessary where computational efficiency is of little consequence. In order to identify this as novel, therefore, we state *'Importantly, identifying the optimum neighbourhood radius for $10^3 - 10^4$ point clouds adds considerable computational cost in processing. As a compromise, the neighbourhood*

*radius of each point in this study is made equal to the distance to the closest point in the reference cloud in Fig. 5a'.* Again, this is clarified in Fig. 3.

- o In a number of instances in this section, it was suggested that elements should be moved into the results section. We describe moved sections below in **Results**. However, we feel that some elements do not constitute 'results' given that they are purely methodological, and are hence better suited to the method section. This applies to the map of optimal search radiuses (Section 3.3); while our use of this technique is new, we do not present any results specifically related to this step. It also applies to the example of the performance of change detection using varying cylinder lengths for the subset of points in Fig. 8 (Section 3.4). While the three different change estimates that we find for this subset are an important justification for varying the cylinder length, we feel that most of this section should remain in the Section 3.4. However, we have opted to move the resulting decrease in LoDs (across the entire point cloud) into Section 4.1.

- o We have divided some of the more lengthy paragraphs (identified in the in text amendments) into smaller paragraphs.

- **Results:** As requested, we have split this section into two, with the first half relating to results of the method and the second to the implications for magnitude-frequency distributions.

- o Section 4.1 provides the results of the filtering, registration, and method of change detection. These have been moved from the methods section as suggested.

- o Section 4.2 remains unchanged.

- **Discussion:** While the content remains broadly unchanged, this has now been split into sections that more accurately reflect the methodological focus of the manuscript.

- **Conclusion:** This more clearly highlights novel conclusions drawn from our experience of processing near-continuous monitoring data. The final paragraph outlines our conclusions relating to the effects of temporal resolution. While a suggested addition was future research for near-continuous processing, we feel that this has been addressed in the discussion. References to previous work have been removed.

- **Figures:** We have ensured that all figure captions begins with a figure 'title', and have shortened the captions for Fig. 9 and Fig. 10 (filter sensitivity analysis), as requested.

- o Fig. 1: more clearly identified lighthouses by shading grey

- o Fig. 3: added a column for software and another for developments from this study. Names, such as waveform deviation, are now consistent with (and within) the rest of the text. We feel that added citations to previous work for each step would add considerable length to the reference list and to the figure/table to which they would be inserted. We feel that the most relevant works have been described and cited in text.

- o Fig. 4 (previously 6): rockfall labelled

- o Fig. 10 (previously 5): (b) reduced to a single curve for clarity both in the plot and in the caption.

  Fig. 12: (b) recued to point data, given that kernel densities (in previous version) simply described the distribution of both variables.

We hope that you feel the above amendments suitably address your suggestions. Please do not hesitate to contact us should you require any further clarity.

Sincerely,

Dr Jack G. Williams

[revised manuscript text omitted]